# Effectiveness of nirmatrelvir/ritonavir in children and adolescents aged 12–17 years following SARS-CoV-2 Omicron infection: A target trial emulation

Carlos K. H. Wong [1,2,3,4,8] ✉, Kristy T. K. Lau[1,8], Ivan C. H. Au[1,5,8], Sophelia H. S. Chan [6], Eric H. Y. Lau [2,7], Benjamin J. Cowling [2,7] & Gabriel M. Leung [2,7]

Currently there is a lack of randomized trial data examining the use of the antiviral nirmatrelvir/ritonavir in paediatric patients with SARS-CoV-2 infection. This target trial emulation study aims to address this gap by evaluating the use of nirmatrelvir/ritonavir in non-hospitalized paediatric patients aged 12–17 years with SARS-CoV-2 Omicron variant infection. Among paediatric patients diagnosed between 16th March 2022 and 5th February 2023, exposure was defined as outpatient nirmatrelvir/ritonavir treatment within 5 days of symptom onset or COVID-19 diagnosis. Primary outcome was 28 day all-cause mortality or all-cause hospitalization, while secondary outcomes were 28 day in-hospital disease progression, 28 day COVID-19-specific hospitalization, multisystem inflammatory syndrome in children (MIS-C), acute liver injury, acute renal failure, and acute respiratory distress syndrome. Overall, 49,378 eligible paediatric patients were included. Nirmatrelvir/ritonavir treatment was associated with reduced 28 day all-cause hospitalization (absolute risk reduction = 0.23%, 95%CI = 0.19%–0.31%; relative risk = 0.66, 95% CI = 0.56–0.71). No events of mortality, in-hospital disease progression, or adverse clinical outcomes were observed among nirmatrelvir/ritonavir users. The findings confirmed the effectiveness of nirmatrelvir/ritonavir in reducing all-cause hospitalization risk among non-hospitalized pediatric patients with SARS-CoV-2 Omicron variant infection.

Paediatric patients with obesity or pre-existing comorbidities such as neurological or cardiac conditions are at an increased risk of severe coronavirus disease 2019 (COVID-19)[1]. An upsurge in the hospitalization rate among COVID-19 patients aged 0–17 years was observed during the period when the SARS-CoV-2 Omicron variant predominated over Delta, especially among unvaccinated individuals[2]. Following the pivotal EPIC-HR trial, the U.S. Food and Drug Administration issued an Emergency Use Authorization (EUA) on 22nd

December 2021 for ritonavir-boosted nirmatrelvir (nirmatrelvir/ritonavir) for treating high-risk adolescents and adults (aged 12 or above) with mild-to-moderate COVID-19[3]; and later on 27th January 2022, the European Medicines Agency recommended Conditional Marketing Authorization for nirmatrelvir/ritonavir to treat high-risk adult patients with COVID-19 who do not require supplemental oxygen[4]. Among adolescents aged 12–17 years who are at high risk of severe COVID-19, nirmatrelvir/ritonavir may be indicated for non-hospitalized patients

with mild-to-moderate disease[5,6]. However, such recommendation is extrapolated from efficacy data of the EPIC-HR trial conducted among adult patients and during the period of SARS-CoV-2 Delta variant predominance[7]. The PANORAMIC trial that investigates the use of nirmatrelvir/ritonavir among community-dwelling COVID-19 patients during the Omicron wave also excludes those aged <18 years[8]; and real-world evidence supporting the use of nirmatrelvir/ritonavir in the paediatric population is scarce[9–11]. In view of the lack of trial data, this retrospective cohort study aims to emulate a hypothetical randomized controlled trial by using cloning methods, to evaluate the clinical effectiveness of nirmatrelvir/ritonavir in non-hospitalized paediatric patients with SARS-CoV-2 Omicron variant infection.

## Results

A total of 50,096 paediatric patients aged 12–17 years had confirmed SARS-CoV-2 infection diagnosis during the study period (Fig. 1). Following the exclusion of those ineligible for current analysis, 49,378 eligible patients in the community setting were identified. Demographics and clinical characteristics of nirmatrelvir/ritonavir users and non-users at the end of the drug initiation period (day 5) before and after the inverse probability of censoring weighting (IPCW) are presented in Supplementary Table 3 and main Table 1, respectively. Most nirmatrelvir/ritonavir users and controls had limited underlying medical conditions, and around 75% were fully vaccinated and/or boosted against COVID-19. None of the nirmatrelvir/ritonavir users had received COVID-19 drug regimens such as molnupiravir, remdesivir or monoclonal antibody therapies during the nirmatrelvir/ritonavir initiation period (i.e., 5 days of grace period), whilst 1 control initiated remdesivir within the first 5 days of index date.

Over a median follow-up of 28 days in both groups, the cumulative incidences of 28 day all-cause hospitalization were 0.45% and 0.68% in the nirmatrelvir/ritonavir and control groups, respectively (Fig. 2). Considering the cause of hospital admission, COVID-19 accounted for 200 of 211 (94.8%) hospitalizations among nirmatrelvir/ritonavir users, and 215 of 332 (64.8%) hospitalizations among controls (Supplementary Table 4). Nirmatrelvir/ritonavir use was associated with a statistically significant reduction in 28 day all-cause hospitalization risk (absolute risk reduction = 0.23%, 95% CI = 0.19–0.31%; relative risk = 0.66, 95%CI = 0.56–0.71). Results from sensitivity analyses were consistent with the main results. (Supplementary Table 5). No events of mortality, in-hospital disease progression, or adverse clinical outcomes were observed among nirmatrelvir/ritonavir users (Supplementary Table 6).

## Discussion

In this highly vaccinated cohort, the cumulative incidences of all-cause hospitalization following SARS-CoV-2 Omicron variant infection in the paediatric population were considerably low, accounting for only 0.45% of nirmatrelvir/ritonavir group and 0.68% of control group. Outpatient nirmatrelvir/ritonavir use was associated with a statistically significant reduction in the all-cause hospitalization risk in the current study. While the risk of severe disease was minimal among our paediatric patients, it should be interpreted in the context of their relatively low risk at baseline and substantial population immunity from adequate vaccination[12].

Our study is to evaluate the real-world effectiveness of nirmatrelvir/ritonavir in non-hospitalized, paediatric COVID-19 patients with a control group, beyond that of feasibility in compassionate use (including only five nirmatrelvir/ritonavir users and 30 matched controls in an inpatient setting)[11]. The cumulative incidence of all-cause hospitalization among our nirmatrelvir/ritonavir users (0.45%) was comparatively <5.7% observed in a US paediatric patient cohort (without a control group) from 1st December 2021–29th September 2022, where 64% of their nirmatrelvir/ritonavir recipients had pre-existing chronic diseases[9]. Another study describing the use of

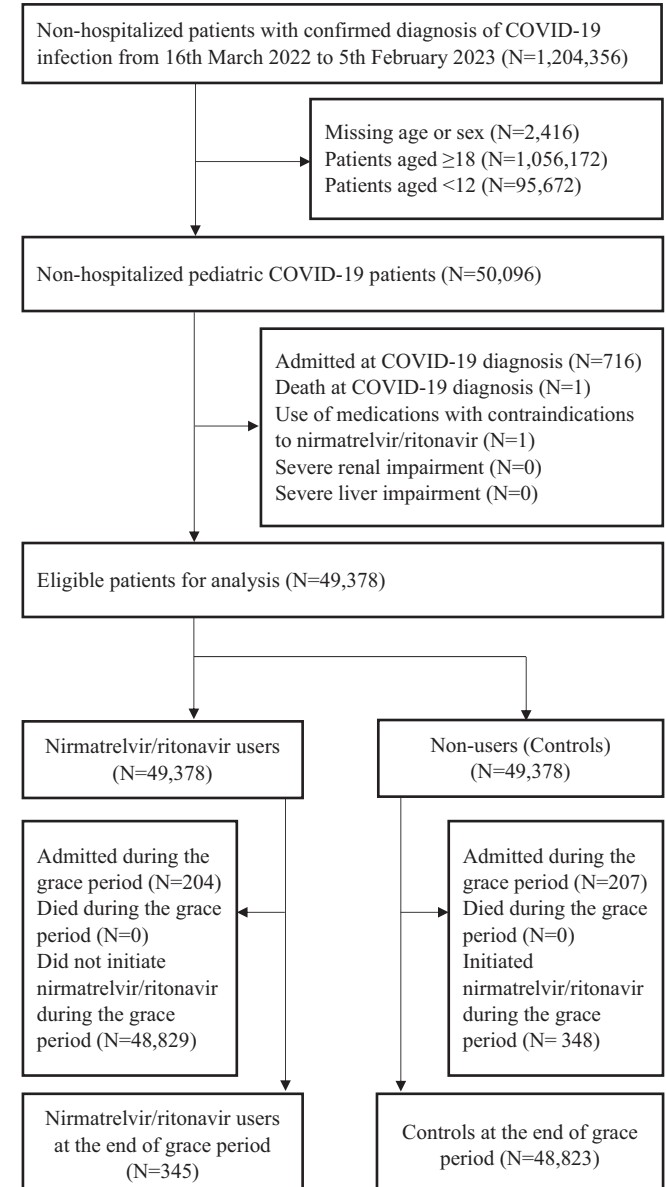

**Fig. 1 | Identification of eligible non-hospitalized paediatric patients with COVID-19 from 16th March 2022 to 5th February 2023 in Hong Kong.** Paediatric patients with confirmed SARS-CoV-2 infection diagnosis during the study period were identified from the data source, stratified by any prescription of outpatient nirmatrelvir/ritonavir treatment. Following the application of study inclusion and exclusion criteria, 49,378 eligible patients were included for analysis, while 345 nirmatrelvir/ritonavir users and 48,823 controls were identified at the end of grace period.

monoclonal antibody and antiviral therapies in paediatric patients with mild-to-moderate COVID-19 found zero hospitalization events during the 7 day follow-up period among 12 nirmatrelvir/ritonavir users (again without a control group)[10].

Considering potential adverse events associated with nirmatrelvir/ritonavir use, Yan et al. have previously reported a transient elevation of liver enzymes after the initiation of this oral antiviral in a treated paediatric patient, which was subsequently recovered following the completion of the 5-day treatment course[11]. Meanwhile, acute liver injury (ALI) was not evident among our nirmatrelvir/ritonavir users. Given no events of other adverse clinical outcomes were observed in our treated patients, nirmatrelvir/ritonavir appeared to be well tolerated by paediatric patients aged 12–17 years.

**Table 1 | Demographics and clinical characteristics of all eligible patients at baseline, nirmatrelvir/ritonavir group and control group at the end of the 5 day grace period after the inverse probability of censoring weighting (IPCW)**

| Characteristics | All eligible patients | | After IPCW at the end of the grace period (Day 5) | | | | |
|---|---|---|---|---|---|---|---|
| | Overall (N = 49,378) | | Nirmatrelvir/ritonavir users (N = 345) | | Controls (N = 48,823) | | SMD |
| | N/Mean | %/SD | N/Mean | %/SD | N/Mean | %/SD | |
| Age, years | 14.4 | 1.7 | 14.5 | 1.6 | 14.4 | 1.7 | 0.03 |
| 12-13 | 17,222 | 34.9% | 114 | 33.1% | 17,032 | 34.9% | 0.04 |
| 14-15 | 16,341 | 33.1% | 120 | 34.9% | 16,145 | 33.1% | |
| 16-17 | 15,815 | 32.0% | 110 | 32.0% | 15,646 | 32.0% | |
| Sex | | | | | | | |
| Male | 26,040 | 52.7% | 196 | 56.8% | 25,750 | 52.7% | 0.09 |
| Female | 23,338 | 47.3% | 149 | 43.2% | 23,073 | 47.3% | |
| SARS-CoV-2 infection period | | | | | | | |
| March 2022 - May 2022 | 8,585 | 17.4% | 68 | 19.6% | 8,508 | 17.4% | 0.07 |
| June 2022 - September 2022 | 25,941 | 52.5% | 183 | 52.9% | 25,659 | 52.6% | |
| October 2022 - January 2023 | 14,852 | 30.1% | 95 | 27.5% | 14,656 | 30.0% | |
| Symptomatic presentation | 27,043 | 54.8% | 188 | 54.5% | 26,723 | 54.7% | 0.01 |
| Pre-existing conditions | | | | | | | |
| Asthma | 7 | 0.0% | 0 | 0.0% | 3 | 0.0% | NA |
| Cancer | 4 | 0.0% | 0 | 0.0% | 1 | 0.0% | NA |
| Cardiac disease | 3 | 0.0% | 0 | 0.0% | 1 | 0.0% | NA |
| Lung disease | 65 | 0.1% | 0 | 0.1% | 42 | 0.1% | 0.04 |
| Mental disease | 12 | 0.0% | 0 | 0.0% | 3 | 0.0% | NA |
| Neurologic disease | 12 | 0.0% | 0 | 0.0% | 5 | 0.0% | NA |
| Obesity | 2 | 0.0% | 0 | 0.0% | 0 | 0.0% | NA |
| Diabetes mellitus | 5 | 0.0% | 0 | 0.0% | 3 | 0.0% | NA |
| Disabilities | 17 | 0.0% | 0 | 0.0% | 3 | 0.0% | NA |
| ADHD | 3 | 0.0% | 0 | 0.0% | 1 | 0.0% | NA |
| Autism | 1 | 0.0% | 0 | 0.0% | 0 | 0.0% | NA |
| Immunocompromised | 40 | 0.1% | 0 | 0.1% | 21 | 0.0% | 0.03 |
| Healthcare utilization | 2,706 | 5.5% | 25 | 7.4% | 2,634 | 5.4% | 0.08 |
| COVID-19 vaccination status[a] | | | | | | | |
| Not fully vaccinated | 12,485 | 25.3% | 89 | 25.8% | 12,362 | 25.3% | 0.03 |
| Fully vaccinated but not boosted | 17,138 | 34.7% | 122 | 35.5% | 16,942 | 34.7% | |
| Boosted | 19,755 | 40.0% | 134 | 38.7% | 19,519 | 40.0% | |

Notes: *ADHD* attention deficit hyperactivity disorder, *SD* standard deviation, *SMD* standardized mean difference, *NA* not applicable.
[a] Fully vaccinated but not boosted patients were defined as those with two doses of BNT162b2 (Comirnaty) or COVID-19 Vaccine (Vero Cell), Inactivated (CoronaVac); boosted patients were defined as those with at least three doses of BNT162b2 (Comirnaty) or COVID-19 Vaccine (Vero Cell), Inactivated (CoronaVac).

This target trial emulation study using territory-wide, retrospective cohort data is timely in illustrating the effectiveness of nirmatrelvir/ritonavir in the paediatric population during the COVID-19 pandemic predominated by SARS-CoV-2 Omicron variant. Despite our attempt of adopting the target trial emulation approach to address the immortal time bias, our study could be limited by potential residual confounding, selection and misclassification biases, ascertainment bias (underreporting of symptoms presented upon diagnosis), absence of treatment records and clinical outcomes of patients engaged with the private healthcare system, and the lack of data on drug compliance among oral antiviral users. While it has been recognized that early initiation of nirmatrelvir/ritonavir (within 5 days of symptom onset) is an important determinant of antiviral effectiveness[13], symptom onset date was only available for 33.9% of our nirmatrelvir/ritonavir users (94.4% of those with symptomatic presentation); and hence, the reduction in all-cause hospitalization risk might have been underestimated, as identifying eligible patients within 5 days of SARS-CoV-2 infection diagnosis has likely included those with nirmatrelvir/ritonavir initiation beyond 5 days of symptom onset. Moreover, clinical notes and information on body weight and body

mass index (BMI) of patients were not available from the anonymous electronic health records to indicate why nirmatrelvir/ritonavir was prescribed to some patients but not others, or if individual patients had refused the drug. While a significant risk reduction in the all-cause hospitalization outcome was observed in the current analysis, the cumulative incidences were relatively low in both treatment and control groups, hence the cost-effectiveness of nirmatrelvir/ritonavir use in the paediatric population will have to be ascertained in other healthcare settings. While the association between oral antiviral use and development of long COVID was not evaluated here, further studies with longer follow-up can explore any protective effect of nirmatrelvir/ritonavir on long COVID among paediatric patients.

In conclusion, this target trial emulation study confirmed the effectiveness of nirmatrelvir/ritonavir in reducing all-cause hospitalization risk among non-hospitalized paediatric patients with SARS-CoV-2 Omicron variant infection. No safety signals were reported following outpatient nirmatrelvir/ritonavir use among paediatric patients aged 12–17 years. Further studies and the ongoing EPIC-Peds trial[14] will provide additional evidence on the clinical efficacy of nirmatrelvir/ritonavir use in children and adolescents with COVID-19, and its safety

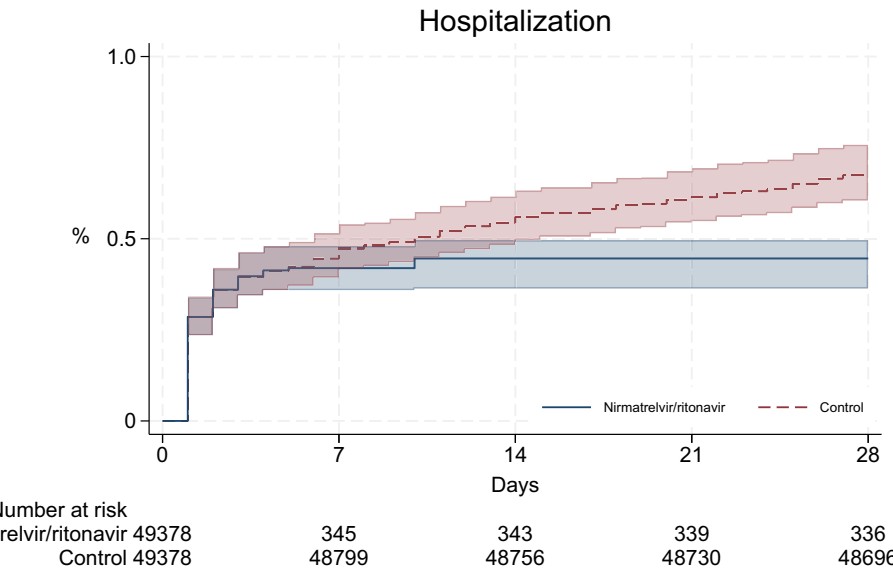

**Fig. 2 | Cumulative incidence of all-cause hospitalization in nirmatrelvir/rito-navir and control groups over the 28 days period.** Line plot represents the cumulative incidence of all-cause hospitalization in nirmatrelvir/ritonavir (blue line) and control groups (red line) over the 28 days follow-up period. Shaded regions represent 95% confidence bands calculated using 500 bootstrap replicates.

Table represents number at risk in nirmatrelvir/ritonavir and control groups after cloning. Over 28 days of follow-up, the cumulative incidences of all-cause hospitalization were 0.45% and 0.68% among nirmatrelvir/ritonavir users and non-users, respectively.

will be confirmed by adverse drug reaction monitoring in the paediatric population.

## Methods

### Study design, data sources, and population

Adopting the target trial emulation framework developed by Hernan et al.[15], this study used data from a territory-wide, retrospective cohort of paediatric patients aged 12–17 years with confirmed SARS-CoV-2 infection from 16th March 2022 (the date when nirmatrelvir/ritonavir was available for prescription locally) to 5th February 2023 in the Hong Kong Special Administrative Region, China. Application of target trial emulation using observational data is aimed at preventing errors and biases, and thus resulting in potentially erroneous causal conclusions[15,16]. Implementation of features such as the choice of the time zero (or the start of follow-up) and random treatment assignment would improve the comparability of the intervention and control groups, allowing more accurate causal inferences on the treatment effects. Specifications of the main components of a hypothetical target trial and our emulated trial using observational data are described in Supplementary Table 1. SARS-CoV-2 infection was confirmed by a positive reverse transcription polymerase chain reaction (RT-PCR) or rapid antigen test (RAT) from the Centre for Health Protection, Department of Health (DH) of the Hong Kong Special Administrative Region. In Hong Kong, all public healthcare (inpatient and outpatient) services are operated by the Hospital Authority (HA). Electronic health records of COVID-19 patients managed under the public healthcare system were retrieved from the HA, and linked COVID-19 vaccination records were obtained from the DH using unique identification numbers. Relevant data extracted from electronic health records included patient demographics, clinical diagnoses, laboratory test results, prescription and dispensing of drugs, hospital admissions, inpatient procedures, and registered deaths. Mortality events were extracted from the Hong Kong Death Registry, capturing both in-hospital deaths and deaths outside hospitals. The database has previously been used in studies evaluating the effectiveness of pharmaceutical treatments against COVID-19 at a population level[17–22].

Indication to use nirmatrelvir/ritonavir in patients aged 12–17 was (1) the initiation of nirmatrelvir/ritonavir within 5 days of symptom

onset or COVID-19 diagnosis, (2) positive RT-PCR or RAT of SARS-CoV-2 infection, and (3) oxygen saturation at room air above 94% (SpO2 > 94%)[23]. However, symptomatic presentation upon diagnosis was not an indication for prescribing nirmatrelvir/ritonavir in patients aged 12–17[23]. Nirmatrelvir/ritonavir users were defined as patients who had received 300 mg nirmatrelvir and 100 mg ritonavir twice daily for 5 days, or 150 mg nirmatrelvir and 100 mg ritonavir twice daily for 5 days among those with moderate renal impairment (estimated glomerular filtration rate [eGFR] ≥ 30 – < 60 mL/min/1.73 m²), according to their drug prescription and dispensing records. Nirmatrelvir/ritonavir treatment was administered orally in all treated patients.

We excluded patients who were aged <12 or >17 years, admitted to hospital or dead on or before the SARS-CoV-2 infection diagnosis, those with severe renal impairment[6] (eGFR <30 mL/min/1.73 m², dialysis, or renal transplantation) or severe liver impairment[6] (cirrhosis, hepatocellular carcinoma, or liver transplantation) at baseline, or drug contraindications to nirmatrelvir/ritonavir (namely amiodarone, carbamazepine, phenobarbital, phenytoin, primidone, St John's wort, apalutamide, enzalutamide, ivosidenib, ivacaftor/lumacaftor, rifampicin, and rifapentine)[24]. Eligibility assessment was performed on the index date (time zero). Among non-hospitalized patients who were eligible for receiving nirmatrelvir/ritonavir, patients were classified into treatment or control groups based on their drug dispensing and prescription records during the first 5 days (grace period) from the index date. Index date was defined as that of SARS-CoV-2 infection diagnosis. Patients were observed from the index date until event occurrence, date of registered death, 28 days since the index date, or 12th February 2023 (end of the follow-up period), whichever came first.

This study was approved by the institutional review board of the University of Hong Kong/Hospital Authority Hong Kong West Cluster (reference no. UW 20-493). Individual patient-informed consent was not required for this retrospective cohort study using anonymized data.

### Study outcomes

The primary outcome was 28 day all-cause mortality or all-cause hospitalization. Secondary outcomes included 28-day in-hospital disease progression (a composite outcome of in-hospital death, invasive

mechanical ventilation, or admission to intensive care unit), 28 day COVID-19-specific hospitalization, multisystem inflammatory syndrome in children (MIS-C) (those who fulfilled the CDC criteria)[25], ALI, acute renal failure, and acute respiratory distress syndrome. Supplementary Table 2 details the definitions of the above clinical outcomes based on International Classification of Diseases, Ninth Revision, Clinical Modification (ICD-9-CM) diagnosis codes and/or laboratory parameters.

## Statistical analyses

Baseline covariates of paediatric patients with COVID-19 included age, sex, date of SARS-CoV-2 infection (March 2022 to May 2022, June 2022 to September 2022, or October 2022 to January 2023), any symptomatic presentation, pre-existing conditions (including asthma, cancer, cardiac disease, lung disease, mental disease, neurologic disease, obesity, diabetes mellitus, disabilities, and immunocompromised status), COVID-19 vaccination status (not fully vaccinated, fully vaccinated but not boosted, or boosted), and healthcare utilization (any inpatient and/or outpatient encounters) over the past year. Immunocompromised status was defined as patients with primary immunodeficiencies, or on active immunosuppressive treatment at baseline or in the past 12 months. Fully vaccinated but not boosted was defined as having received two doses of BNT162b2 (Comirnaty) or COVID-19 Vaccine (Vero Cell), Inactivated (CoronaVac)[26]. Boosted was defined as having received at least three doses of BNT162b2 or CoronaVac.

Standardized mean differences (SMDs) of characteristics of nirmatrelvir/ritonavir group and control group at the end of the 5 day grace period before and after the IPCW, and were interpreted as balanced when SMDs < 0.1[27].

By applying the cloning method[15,27] we created two clones of all eligible patients, and assigned each of the two clones to either nirmatrelvir/ritonavir group or control group on the index date (time zero). The cloning method avoided immortal time bias due to defining nirmatrelvir/ritonavir initiation among those survived and non-hospitalized paediatric patients[15,27]. The start of follow-up (time zero) and nirmatrelvir/ritonavir initiation did not often happen on the same day for some patients who survived long enough to initiate treatment with nirmatrelvir/ritonavir some days after the start of follow-up[28]. Immortal time bias occurred when time period between the start of follow-up and nirmatrelvir/ritonavir initiation was misclassified or excluded by the analysis. In the current study, clones were then censored at the time when they deviated from the treatment protocol of each group during a grace period of 5 days since the index date, i.e. clones who received nirmatrelvir/ritonavir within 5 days of the index date were censored at their time of nirmatrelvir/ritonavir initiation in the control group, while clones who did not receive nirmatrelvir/ritonavir within 5 days were censored at day 5 in the treated group. The 5 days of grace period was based on the EUA stating that nirmatrelvir/ritonavir should be initiated as soon as possible after COVID-19 diagnosis and within 5 days of symptom onset[6]. Inverse probability of censoring weighting (IPCW) at each day during the nirmatrelvir/ritonavir initiation period (i.e. 5 days of grace period)(Supplementary Fig. 1) was used to address the selection bias over time introduced by this informative censoring. IPCW was the inverse of clones' probabilities of remaining uncensored by baseline covariates described above (continuous form of age, sex, date of SARS-CoV-2 infection, symptomatic presentation, pre-existing conditions, COVID-19 vaccination status, healthcare utilization over the past year of patients, and interaction terms for covariate balance[27]), while those probabilities were estimated using a pooled logistic regression model. The cumulative incidence functions through a weighted non-parametric Kaplan-Meier estimator[29] were used to estimate the absolute risk reduction by 28 days due to nirmatrelvir/ritonavir, and relative risk at 28 days (i.e. ratio of cumulative incidence at 28 days of nirmatrelvir/ritonavir group over that of control group), with 95% confidence intervals (CIs) constructed using a nonparametric bootstrap of 500 samples.

To test the robustness of main results, sensitivity analyses were performed by (1) truncating the IPCW at the 1st and 99th percentiles to minimize the impact of extreme weights on the results; and (2) extending the follow-up duration till the end of observational period.

All statistical analyses were performed using Stata/MP (version 18). All significance tests were two-tailed, and $p$-value < 0.05 was considered statistically significant.

## Ethics approval

This study was approved by the institutional review board of the University of Hong Kong/Hospital Authority Hong Kong West Cluster (reference no. UW 20-493). Individual patient-informed consent was not required for this retrospective cohort study using anonymized data.

## Data access, responsibility and analysis

I.C.H.A and C.K.H.W. had full access to all the data in the study and takes responsibility for the integrity of the data and the accuracy of the data analysis.

## Reporting summary

Further information on research design is available in the Nature Portfolio Reporting Summary linked to this article.

## Data availability

The clinical outcome data and vaccination records were extracted from the Hospital Authority database in Hong Kong and data on confirmed cases of SARS-CoV-2 infection were extracted from the eSARS data provided by the Centre for Health Protection (Department of Health, The Government of the Hong Kong Special Administrative Region). The data custodians (the Hospital Authority and the Department of Health) provided the underlying individual patient data to The University of Hong Kong for the purpose of performing scientific research for the study. Restrictions apply to the availability of these data, which were used under licence of the Hospital Authority and the Department of Health for this study. The authors cannot transmit or release the data, in whole or in part in whatever form or media, or to any other parties or place outside Hong Kong; and the authors fully comply with the duties under the laws of Hong Kong relating to the protection of personal data including those under the Personal Data (Privacy) Ordinance and its principles in all aspects.

## Code availability

The code used for this study is publicly available on GitHub (https://github.com/IvanAuHKU/COVID-nirmatrelvir-ritonavir-paediatric)[30].

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

## Acknowledgements

This study was supported by the Health and Medical Research Fund (reference number: COVID190210), Health Bureau, Government of Hong Kong Special Administrative Region, China; AIR@InnoHK administered by Innovation and Technology Commission, Government of Hong Kong Special Administrative Region, China. The funding source had no involvement in the study design; in the collection, analysis and interpretation of data; in the writing of the report; or in the decision to submit the article for publication.

## Author contributions

C.K.H.W. and I.C.H.A. designed the study, wrote the manuscript, contributed to the interpretation of the analysis, conducted analysis, revised the manuscript, and attests that all listed authors meet authorship criteria and that no others meeting the criteria have been omitted. K.T.K.L. reviewed the literature, wrote the manuscript, contributed to the interpretation of the analysis, and revised the manuscript. S.H.S.C. contributed to the interpretation of the analysis. C.K.H.W., I.C.H.A., and E.H.Y.L. accessed and verified the underlying data. S.H.S.C, E.H.Y.L, B.J.C., and G.M.L. reviewed and revised the manuscript.

## Competing interests

C.K.H.W. reports the receipt of General Research Fund, Research Grant Council, Government of Hong Kong SAR; EuroQol Research Foundation; AstraZeneca and Boehringer Ingelheim, all outside the submitted work. B.J.C. reports honoraria from AstraZeneca, Fosun Pharma, GlaxoSmithKline, Haleon, Moderna, Pfizer, Roche, and Sanofi Pasteur. B.J.C. has provided scientific advice to Pfizer and AstraZeneca on issues related to COVID-19 disease burden and vaccine effectiveness. He has not provided scientific advice to either company related to COVID-19 antiviral effectiveness, and he has not received any funding from Pfizer or AstraZeneca for any research on antiviral effectiveness including the current work. All other authors declare no competing interests.

## Additional information

[1]Department of Pharmacology and Pharmacy, LKS Faculty of Medicine, The University of Hong Kong, Hong Kong SAR, China. [2]Laboratory of Data Discovery for Health (D24H), Hong Kong SAR, China. [3]Department of Family Medicine and Primary Care, School of Clinical Medicine, LKS Faculty of Medicine, The University of Hong Kong, Hong Kong SAR, China. [4]Department of Infectious Disease Epidemiology & Dynamics, Faculty of Epidemiology and Population Health, London School of Hygiene and Tropical Medicine, London, UK. [5]School of Public Health, LKS Faculty of Medicine, The University of Hong Kong, Hong Kong SAR, China. [6]Department of Paediatrics and Adolescent Medicine, School of Clinical Medicine, LKS Faculty of Medicine, The University of Hong Kong, Hong Kong SAR, China. [7]WHO Collaborating Centre for Infectious Disease Epidemiology and Control, School of Public Health, LKS Faculty of Medicine, The University of Hong Kong, Hong Kong SAR, China. [8]These authors contributed equally: Carlos K. H. Wong, Kristy T. K. Lau, Ivan C. H. Au. ✉e-mail: carlosho@hku.hk

