## [Peer Review File · Nature Communications]

Effectiveness of nirmatrelvir/ritonavir in children and adolescents aged 12-17 years following SARS-CoV-2 Omicron infection: A target trial emulationReviewers' Comments:

Reviewer #1:

Remarks to the Author:

Wong et al conducted an observational study among non-hospitalized pediatric patients aged 12-17 years diagnosed with COVID-19 in Hong Kong and compared 28-day outcomes, including all-cause mortality or hospitalization, between adolescents who received nirmatrelvir-ritonavir versus no COVID-19 treatment.

Major comments

1. Are there additional eligibility criteria which should be applied here (Figure 1, Supplementary Table 1)? There is no mention of minimum weight (40 kg per FDA) and risk factors (i.e., comorbidities) for progression to severe COVID-19.
2. The index date (time zero) should be the date at which eligibility is assessed, treatment arm assigned, and follow-up begins (not just start of follow-up as is mentioned in lines 265-66 in the Methods). These elements will not be aligned if different definitions for the index date are used (in this study, the earliest of infection diagnosis, symptom onset, or antiviral initiation, as mentioned in lines 302-04). Further, if symptom ascertainment was different between comparator groups (as suggested by Supplementary Table 3 where 55% of controls had a documented symptomatic presentation but only 36% of nirmatrelvir-ritonavir users), then there will be an imbalance in the assignment of the index date. These choices affect inferences drawn about the effectiveness of nirmatrelvir-ritonavir, as shown in Supplementary Table 6, where no ARR is observed when patients whose assigned index date was the treatment initiation date when it occurred before diagnosis/symptom onset date were excluded. I suggest using the same index date throughout (e.g., diagnosis date for both groups and excluding anyone treated before diagnosis).
3. Please explain why both PS matching and cloning were used. With the cloning approach, additional PS matching is not needed because confounding at baseline is addressed through this method. With cloning alone, you also don't need to divide groups into ever/never treated as the authors did for the PS models. I wonder if the cloning alone method should be presented as the main approach rather than as a sensitivity analysis.
4. The authors should provide more details on how the cloning was executed, both for the cloning alone approach as well as the approach combining PS matching (if the latter is kept in the manuscript):
 - a. Cloning alone should not include ever/never treated as this can be addressed through deviation from the assigned arm.
 - b. Who was included? Denominator data should be presented for Supplementary Table 6.
 - c. What was included in the IPCW models?
 - d. Please show balance of covariates unweighted and with IPCW (either at the end of the grace period or average during this time).
 - e. Why are IPCWs calculated for days >5 if persons are only eligible for treatment on days 1-5?
5. Given different sensitivity analysis estimates in Supplementary Table 6 and the potential for drawing different inferences, please include some comment on this in the Discussion.

Minor comments

1. Table 1 implies that only a third of treated children had symptoms. Nirmatrelvir-ritonavir is only intended for use in symptomatic persons, was symptom ascertainment incomplete?
2. Please explain why a large number of secondary outcomes was considered.

Reviewer #2:

Remarks to the Author:

I co-reviewed this manuscript with one of the reviewers who provided the listed reports. This is part of the Nature Communications initiative to facilitate training in peer review and to provide appropriate

recognition for Early Career Researchers who co-review manuscripts.

Reviewer #3:

Remarks to the Author:

****Summary****

The authors evaluated the real-world effectiveness of nirmatrelvir/ritonavir among non-hospitalized, pediatric COVID-19 patients aged 12-17 years in Hong Kong during March 2022 - Feb 2023 when the Omicron variant was predominant. They found that nirmatrelvir/ritonavir was effective in reducing all-cause hospitalization risk among this population. This study provides very important data for this age group (12-17 years) that has often been excluded from previous trials.

A few fundamental considerations and some more detailed comments/questions are included in my review below. I believe that the authors should be able to address the majority of these, and pending adequate revisions, I would accept this manuscript.

****Major comments -- concerns that should be addressed (/ rebutted) before publication****

1. The definition of the index date (SARS-CoV-2 infection diagnosis, symptom onset, or nirmatrelvir/ritonavir initiation, whichever occurred earlier) was a little confusing to me. Why was it possible for nirmatrelvir/ritonavir initiation to happen before the onset of symptom or SARS-CoV-2 infection diagnosis? It makes sense that, in the target trial, the index date does not rely on the initiation of nirmatrelvir/ritonavir, as described in the Supplementary Table 1. The authors excluded the initiation of More explanations would be helpful for
2. Relatedly, in one of the sensitivity analyses in which the authors excluded patients whose index date was defined as that of nirmatrelvir/ritonavir initiation when nirmatrelvir/ritonavir initiation date was earlier than that of COVID-19 diagnosis and/or symptom onset, it seems the reduction in the risk of 28-day all-cause hospitalization disappeared, as shown in Supplementary Table 6 (Absolute risk reduction: -0.04%). The authors say that "results from sensitivity analyses on the all-cause hospitalization outcome were broadly consistent with the main results," but I think further explanations and discussions would be helpful for this particular sensitivity analysis and result.
3. The primary outcome was 28-day all-cause mortality or all-cause hospitalization. What happens if we use 28-day COVID-19-specific hospitalizations as an outcome?
4. Could the authors add 95% CIs to the estimated cumulative risks in Figure 2?
5. Regarding the baseline covariates, how were these variables treated in the Cox PH model for censor weight calculation? Was age numeric or categorical? What about the date of SARS-CoV-2 infection? Was it by month or date? Was it categorical or slines?

****Minor comments -- Addition general comments / concerns to be addressed at the authors' discretion****

1. Is a nonparametric bootstrap of 100 bootstrap samples enough to calculate the 95% CI?
2. I assume COVID-19 vaccination status was included as a categorical variable in the Cox PH model. Would the timing of these doses matter?

REVIEWER COMMENTS

Reviewer #1 (Remarks to the Author):

Wong et al conducted an observational study among non-hospitalized pediatric patients aged 12-17 years diagnosed with COVID-19 in Hong Kong and compared 28-day outcomes, including all-cause mortality or hospitalization, between adolescents who received nirmatrelvir-ritonavir versus no COVID-19 treatment.

Major comments

1. Are there additional eligibility criteria which should be applied here (Figure 1, Supplementary Table 1)? There is no mention of minimum weight (40 kg per FDA) and risk factors (i.e., comorbidities) for progression to severe COVID-19.

Response 1.1:

Nirmatrelvir/ritonavir was not authorized for use in pediatric patients younger than 12 years of age or weighting less than 40 kg, according to FDA emergency use authorization (EUA) for nirmatrelvir/ritonavir. Unfortunately information on body weight and body mass index (BMI) of patients as well as clinical notes were not available from our data source, and thus were not part of our eligibility criteria for this analysis. We have indicated in the limitation of our revised manuscript (page 6 lines 140-141).

Reference:

US Food and Drug Administration. Fact Sheet for Healthcare Providers: Emergency Use Authorization for Paxlovid, <<https://www.fda.gov/media/155050/download>> (2023).

2. The index date (time zero) should be the date at which eligibility is assessed, treatment arm assigned, and follow-up begins (not just start of follow-up as is mentioned in lines 265-66 in the Methods). These elements will not be aligned if different definitions for the index date are used (in this study, the earliest of infection diagnosis, symptom onset, or antiviral initiation, as mentioned in lines 302-04). Further, if symptom ascertainment was different between comparator groups (as suggested by Supplementary Table 3 where 55% of controls had a documented symptomatic presentation but only 36% of nirmatrelvir-ritonavir users), then there will be an imbalance in the assignment of the index date. These choices affect

inferences drawn about the effectiveness of nirmatrelvir-ritonavir, as shown in Supplementary Table 6, where no ARR is observed when patients whose assigned index date was the treatment initiation date when it occurred before diagnosis/symptom onset date were excluded. I suggest using the same index date throughout (e.g., diagnosis date for both groups and excluding anyone treated before diagnosis).

Response 1.2:

Nirmatrelvir/ritonavir was only indicated when patients were diagnosed with SARS-CoV-2 through PCR or RAT testing. The initiation of nirmatrelvir/ritonavir should not be listed as the definition of the index date, so we have corrected the eligibility criteria in the target trial emulation specification (Supplementary Table 1). For synchronicity of time zero (index date) for all eligible patients, we have standardized the use of SARS-CoV-2 infection diagnosis date as the index date.

Correspondingly, 136 control patients who reported symptoms prior to the hospitalization were admitted before diagnosis (new index date) and thus were excluded from this updated analysis. Two control patients became nirmatrelvir/ritonavir users who initiated their nirmatrelvir/ritonavir within 5 days from the new index date. Overall, owing to the change of index date definition and eligibility criteria (See updated Figure 1), the updated analysis reached the same conclusion as the original analysis.

As we have now standardized the use of SARS-CoV-2 diagnosis date as the index date, the sensitivity analysis “Excluding patients whose index date was defined as that of nirmatrelvir/ritonavir initiation when nirmatrelvir/ritonavir initiation date was earlier than that of COVID-19 diagnosis and/or symptom onset” is removed.

3. Please explain why both PS matching and cloning were used. With the cloning approach, additional PS matching is not needed because confounding at baseline is addressed through this method. With cloning alone, you also don't need to divide groups into ever/never treated as the authors did for the PS models. I wonder if the cloning alone method should be presented as the main approach rather than as a sensitivity analysis.

Response 1.3:

The propensity-score matching was intended for mitigating the residual confounding issue due to imbalanced baseline characteristics in two groups within the traditional cohort analytic framework. Within the target trial emulation framework, patients' baseline characteristics in two groups (on time zero) were identical after cloning, so the covariate balance by propensity-score matching was not necessary. We have now removed 1:10 propensity-score matching, and adopted 'cloning alone' approach as main analysis which demonstrated the reduced risks of primary outcome associated with the nirmatrelvir/ritonavir use.

4. The authors should provide more details on how the cloning was executed, both for the cloning alone approach as well as the approach combining PS matching (if the latter is kept in the manuscript):

- a. Cloning alone should not include ever/never treated as this can be addressed through deviation from the assigned arm.*
- b. Who was included? Denominator data should be presented for Supplementary Table 6.*
- c. What was included in the IPCW models?*
- d. Please show balance of covariates unweighted and with IPCW (either at the end of the grace period or average during this time).*
- e. Why are IPCWs calculated for days >5 if persons are only eligible for treatment on days 1-5?*

Response 1.4:

- a. We agree with the reviewer that ever/never treated should not be included in the cloning alone approach, and we have now used cloning alone method as the main approach. In the cloning alone approach, all eligible patients were cloned and replicated in both the treatment and control arms, and then censored if they deviated from the treatment strategy for the respective assigned arm. Information on ever/never treated are no longer applicable for the 'cloning alone' approach.
- b. All 49,378 eligible patients were included in both the treatment and control groups after cloning in the main and sensitivity analyses (Supplementary Table 5).
- c. The IPCW models included all baseline covariates (continuous form of age, sex, date of SARS-CoV-2 infection, any symptomatic presentation, pre-existing conditions, COVID-19 vaccination status, and healthcare utilization over the past year) and interaction terms for covariate balance¹. We have now clarified this in the Methods section of our revised manuscript (page 17 lines 353-357).

d. The patients' covariates (and their balance in term of SMDs) at the end of the grace period (Day 5) before and after the IPCW have been presented in new Supplementary Table 3 and new Table 1 now, respectively. Covariate balance between the two groups at the end of grace period was achieved after applying IPCW, with all SMDs <0.1.

e. We have applied the same approach as did in prior target trial emulation studies^{2,3}. As there was no more informative censoring after the 5-day grace period, the probability of being censored was 0 and the daily weight was 1 for days >5 in both groups. The estimated IPCW, which was the cumulative product of daily weights since day 1, stayed constant from days >5 onwards. This has now been elaborated in Supplementary Figure 1.

References:

1. Maringe C, Benitez Majano S, Exarchakou A, et al. Reflection on modern methods: trial emulation in the presence of immortal-time bias. Assessing the benefit of major surgery for elderly lung cancer patients using observational data. *International Journal of Epidemiology*. 2020;49(5):1719-1729.

2. Hernan MA, Brumback B, Robins JM. Marginal structural models to estimate the causal effect of zidovudine on the survival of HIV-positive men. *Epidemiology*. 2000; 11(5): 561-70.

3. Emilsson L, García-Albéniz X, Logan RW, Caniglia EC, Kalager M, Hernán MA. Examining Bias in Studies of Statin Treatment and Survival in Patients With Cancer. *JAMA Oncology*. 2018;4(1):63-70.

5. Given different sensitivity analysis estimates in Supplementary Table 6 and the potential for drawing different inferences, please include some comment on this in the Discussion.

Response 1.5:

As we are now using the SARS-CoV-2 diagnosis date as the index date, the sensitivity analysis “Excluding patients whose index date was defined as that of nirmatrelvir/ritonavir initiation when nirmatrelvir/ritonavir initiation date was earlier than that of COVID-19 diagnosis and/or symptom onset” is removed.

Minor comments

1. Table 1 implies that only a third of treated children had symptoms. Nirmatrelvir-ritonavir is only intended for use in symptomatic persons, was symptom ascertainment incomplete?

Response 1.6: Under-ascertainment and underreporting in symptoms presented upon SARS-CoV-2 diagnosis is possible, and could introduce potential bias to our findings. We have added this point to the limitation section of the revised manuscript (page 6 line 131).

However, symptomatic presentation was not an indication for prescribing nirmatrelvir/ritonavir according to the Hospital Authority clinical management guideline for pediatric patients with SARS-CoV-2 infection. That explained why our cohort included nirmatrelvir/ritonavir users with and without symptomatic presentation upon diagnosis, as demonstrated in Table 1.

Reference:

HA Central Committee on Infectious Diseases and Emergency Response (CCIDER). (2022).

2. Please explain why a large number of secondary outcomes was considered.

Response 1.7:

We have reduced the number of secondary outcomes considered, and presented a minimum set of secondary outcomes, including 28-day COVID-19-specific hospitalization, 28-day in-hospital disease progression (composite outcome of in-hospital death, invasive mechanical ventilation, and intensive care unit admission), multisystem inflammatory syndrome in children (MIS-C), acute liver injury, acute renal failure, and acute respiratory distress syndrome. They were considered as outcomes of children and adolescents during the acute phase within 28 days of SARS-CoV-2 infection in previous studies¹⁻³. This has now been updated in Supplementary Table 1, Supplementary Table 6, and the Methods section of our revised manuscript (page 15 lines 310-315).

References:

1. Feldstein LR, Tenforde MW, Friedman KG, et al. Characteristics and Outcomes of US Children and Adolescents With Multisystem Inflammatory Syndrome in Children (MIS-C) Compared With Severe Acute COVID-19. *JAMA*. 2021; 325(11):1074-1087.
2. Yousaf AR, Cortese MM, Taylor AW, et al. Reported cases of multisystem inflammatory syndrome in children aged 12–20 years in the USA who received a COVID-19 vaccine, December, 2020, through August, 2021: a surveillance investigation. *The Lancet Child & Adolescent Health*. 2022; 6(5): 303-12.

3. Ward JL, Harwood R, Kenny S, et al. Pediatric Hospitalizations and ICU Admissions Due to COVID-19 and Pediatric Inflammatory Multisystem Syndrome Temporally Associated With SARS-CoV-2 in England. *JAMA Pediatrics*. 2023;177(9):947-955.

Reviewer #2 (Remarks to the Author):

Response 2:

We sincerely appreciate the reviewers for your time to offer invaluable insights on our paper.

Reviewer #3 (Remarks to the Author):

****Summary****

The authors evaluated the real-world effectiveness of nirmatrelvir/ritonavir among non-hospitalized, pediatric COVID-19 patients aged 12-17 years in Hong Kong during March 2022 - Feb 2023 when the Omicron variant was predominant. They found that nirmatrelvir/ritonavir was effective in reducing all-cause hospitalization risk among this population. This study provides very important data for this age group (12-17 years) that has often been excluded from previous trials.

A few fundamental considerations and some more detailed comments/questions are included in my review below. I believe that the authors should be able to address the majority of these, and pending adequate revisions, I would accept this manuscript.

****Major comments -- concerns that should be addressed (/ rebutted) before publication****

1. The definition of the index date (SARS-CoV-2 infection diagnosis, symptom onset, or nirmatrelvir/ritonavir initiation, whichever occurred earlier) was a little confusing to me. Why was it possible for nirmatrelvir/ritonavir initiation to happen before the onset of symptom or SARS-CoV-2 infection diagnosis? It makes sense that, in the target trial, the

index date does not rely on the initiation of nirmatrelvir/ritonavir, as described in the Supplementary Table 1. The authors excluded the initiation of More explanations would be helpful for

Response 3.1:

Nirmatrelvir/ritonavir was only indicated when patients were diagnosed with SARS-CoV-2 through PCR or RAT testing. The initiation of nirmatrelvir/ritonavir should not be listed as the definition of the index date, so we have corrected this eligibility criteria in the target trial emulation specification (Supplementary Table 1). For synchronicity of time zero (index date) for all eligible patients, we have standardized the use of SARS-CoV-2 infection diagnosis date as the index date. Correspondingly, 136 control patients who reported symptoms prior to the hospitalization were admitted before diagnosis (new index date) and thus were excluded from this updated analysis. Two control patients became nirmatrelvir/ritonavir users who initiated nirmatrelvir/ritonavir within 5 days from the new index date.

Overall, owing to the change of index date definition and eligibility criteria (See updated Figure 1), the updated analysis reached the same conclusion as the original analysis.

2. Relatedly, in one of the sensitivity analyses in which the authors excluded patients whose index date was defined as that of nirmatrelvir/ritonavir initiation when nirmatrelvir/ritonavir initiation date was earlier than that of COVID-19 diagnosis and/or symptom onset, it seems the reduction in the risk of 28-day all-cause hospitalization disappeared, as shown in Supplementary Table 6 (Absolute risk reduction: -0.04%). The authors say that “results from sensitivity analyses on the all-cause hospitalization outcome were broadly consistent with the main results,” but I think further explanations and discussions would be helpful for this particular sensitivity analysis and result.

Response 3.2:

As we have now standardized the use of SARS-CoV-2 diagnosis date as the index date, the sensitivity analysis “Excluding patients whose index date was defined as that of nirmatrelvir/ritonavir initiation when nirmatrelvir/ritonavir initiation date was earlier than that of COVID-19 diagnosis and/or symptom onset” is removed.

3. *The primary outcome was 28-day all-cause mortality or all-cause hospitalization. What happens if we use 28-day COVID-19-specific hospitalizations as an outcome?*

Response 3.3:

We have now added the 28-day COVID-19-specific hospitalizations as one of the secondary outcomes, and revised the corresponding sentences under the Method section (page 15 lines 311-315). The relative risk was 0.91 (95% CI 0.86-0.94), showing the same conclusion of reduced risk associated with nirmatrelvir/ritonavir use.

4. *Could the authors add 95% CIs to the estimated cumulative risks in Figure 2?*

Response 3.4:

We have now added the 95% bands to the estimated cumulative incidence in Figure 2. The 95% bands were calculated using the 500 bootstrap replicates.

5. *Regarding the baseline covariates, how were these variables treated in the Cox PH model for censor weight calculation? Was age numeric or categorical? What about the date of SARS-CoV-2 infection? Was it by month or date? Was it categorical or splines?*

Response 3.5 Pooled logistic regression model is more appropriate to model censoring events in the treatment group, and widely used for IPCW estimation in prior target trial emulation studies¹⁻⁴. We have now changed the IPCW model to pooled logistic regression model. The baseline covariates included in the pooled logistic regression models were age, sex, date of SARS-CoV-2 infection, symptomatic presentation, pre-existing conditions, COVID-19 vaccination status, healthcare utilization over the past year of patients, and interaction terms for covariate balance⁵. In particular, age was expressed in numeric continuous form, and date of SARS-CoV-2 infection was expressed in categorical form as shown in Table 1. This has been elaborated under the Method section (page 17 lines 353-357).

References:

1. Emilsson L, García-Albéniz X, Logan RW, Caniglia EC, Kalager M, Hernán MA. Examining Bias in Studies of Statin Treatment and Survival in Patients With Cancer. *JAMA Oncology*. 2018;4(1):63-70.

2. Madenci AL, Wanis KN, Cooper Z, et al. Strengthening Health Services Research Using Target Trial Emulation: An Application to Volume-Outcomes Studies. *American Journal of Epidemiology*. 2021;190(11):2453-2460.
3. Trevisi L, Hernán MA, Mitnick CD, et al. Effectiveness of Bedaquiline Use beyond Six Months in Patients with Multidrug-Resistant Tuberculosis. *American Journal of Respiratory and Critical Care Medicine*. 2023;207(11):1525-1532.
4. Rein SM, Lodi S, Logan RW, et al. Integrase strand-transfer inhibitor use and cardiovascular events in adults with HIV: an emulation of target trials in the HIV-CAUSAL Collaboration and the Antiretroviral Therapy Cohort Collaboration. *Lancet HIV*. 2023;10(11):e723-e732.
5. Maringe C, Benitez Majano S, Exarchakou A, et al. Reflection on modern methods: trial emulation in the presence of immortal-time bias. Assessing the benefit of major surgery for elderly lung cancer patients using observational data. *International Journal of Epidemiology*. 2020;49(5):1719-1729.

Minor comments -- Addition general comments / concerns to be addressed at the authors' discretion

1. Is a nonparametric bootstrap of 100 bootstrap samples enough to calculate the 95% CI?

Response 3.6: Previous study¹ proposed stopping criteria which the threshold of the number of samples (or replicates) should be enough, and recommended stopping criteria to be typically 100-500 bootstrapping samples. We have now increased the bootstrapping samples to 500, and made corresponding changes under the Statistical Analysis subsection (page 17 lines 361-362).

Reference:

1. Pattengale ND, Alipour M, Bininda-Emonds OR, Moret BM, Stamatakis A. How many bootstrap replicates are necessary?. *Journal of Computational Biology*. 2010;17(3):337-354

2. I assume COVID-19 vaccination status was included as a categorical variable in the Cox PH model. Would the timing of these doses matter?

Response 3.7: We have included COVID-19 vaccination status in the pooled logistic regression models for the development of IPCW models. However, timing of last doses was not accounted for in the IPCW model because a small proportion of our cohort (17.3%) has not yet vaccinated. At the end of grace period (day 5) after IPCW, the proportion of patients without any vaccination was 16.7% for nirmatrelvir/ritonavir group and 17.3% for control group with SMD of 0.02; while the mean duration of the last COVID-19 vaccine dose was 9.4 months (SD 4.5) for nirmatrelvir/ritonavir group and 9.5 months (SD 4.1) for control group with SMD of 0.02. Similar mean duration of last doses between the two groups would not imply confounding by timing of doses, and thereby would not affect the conclusion.

We sincerely hope that you will consider this revised manuscript favourably. Should there be further corrections or clarifications needed, we are more than happy to make the necessary changes and provide additional information. We look forward to hearing from you soon.

Yours sincerely,

Carlos K.H. Wong, PhD

Department of Family Medicine and Primary Care, School of Clinical Medicine, Li Ka Shing
Faculty of Medicine, The University of Hong Kong

Rm 1-01, 1/F, Jockey Club Building for Interdisciplinary Research, 5 Sassoon Road,

Pokfulam, Hong Kong, China

Email: carlosho@hku.hk

Reviewers' Comments:

Reviewer #1:

Remarks to the Author:

Thank you for your responses and revisions. With regard to response 1.6, the clarification that symptomatic presentation was not an indication for prescribing nirmatrelvir/ritonavir in your practice setting is important context to provide to readers. This is particularly true since in many settings, symptomatic presentation is a requirement for treatment. This information will help interpret study results and generalizability accordingly.

Reviewer #2:

Remarks to the Author:

Reviewer #3:

Remarks to the Author:

Thank you for addressing my comments. I only have one minor comment, regarding the following Response 3.3.

"Response 3.3: We have now added the 28-day COVID-19-specific hospitalizations as one of the secondary outcomes, and...."

Where are these updates reflected in the revised manuscript? Sorry I missed it, but I could not find it. It would be helpful if the authors could point reviewers to the lines they made changes in the updated manuscript. I think this would be useful information and it would be great if the authors can include it in the main text.

REVIEWER COMMENTS

Reviewer #1 (Remarks to the Author):

Thank you for your responses and revisions. With regard to response 1.6, the clarification that symptomatic presentation was not an indication for prescribing nirmatrelvir/ritonavir in your practice setting is important context to provide to readers. This is particularly true since in many settings, symptomatic presentation is a requirement for treatment. This information will help interpret study results and generalizability accordingly.

Response 1.1:

Thank you for your suggestions. We have clarified that symptomatic presentation was not an indication for prescribing nirmatrelvir/ritonavir in the practice setting in our study. The clarification has been included in the Methods section of our revised manuscript (page 8, lines 185-186).

Reviewer #2 (Remarks to the Author):

Response 2:

We sincerely appreciate the reviewers for your time to offer invaluable insights on our paper.

Reviewer #3 (Remarks to the Author):

Thank you for addressing my comments. I only have one minor comment, regarding the following Response 3.3.

"Response 3.3: We have now added the 28-day COVID-19-specific hospitalizations as one of the secondary outcomes, and...."

Where are these updates reflected in the revised manuscript? Sorry I missed it, but I could not find it. It would be helpful if the authors could point reviewers to the lines they made changes in the updated manuscript. I think this would be useful information and it would be great if the authors can include it in the main text.

Response 3.1: We have included the 28-day COVID-19-specific hospitalizations as one of the secondary outcomes in the Methods section of our revised manuscript (page 10, lines 214-215) and in Supplementary Table 1. Result for the 28-day COVID-19-specific hospitalizations has been included in Supplementary Table 6. Relative risk for 28-day COVID-19-specific hospitalization was 0.91 (95% CI 0.85-0.94).

Should there be further corrections or clarifications needed, we are more than happy to make the necessary changes and provide additional information. We look forward to hearing from you soon.

Yours sincerely,

Carlos K.H. Wong, PhD

Department of Family Medicine and Primary Care, School of Clinical Medicine, Li Ka Shing Faculty of Medicine, The University of Hong Kong

Rm 1-01, 1/F, Jockey Club Building for Interdisciplinary Research, 5 Sassoon Road, Pokfulam, Hong Kong, China

Email: carlosho@hku.hk